# Mice Lacking the Systemic Vitamin A Receptor RBPR2 Show Decreased Ocular Retinoids and Loss of Visual Function

**DOI:** 10.3390/nu14122371

**Published:** 2022-06-08

**Authors:** Rakesh Radhakrishnan, Matthias Leung, Heidi Roehrich, Stephen Walterhouse, Altaf A. Kondkar, Wayne Fitzgibbon, Manas R. Biswal, Glenn P. Lobo

**Affiliations:** 1Department of Ophthalmology and Visual Neurosciences, University of Minnesota, Lions Research Building, 2001 6th Street SE, Minneapolis, MN 55455, USA; rakeshr@umn.edu (R.R.); leung132@umn.edu (M.L.); rohri002@umn.edu (H.R.); 2Department of Medicine, Medical University of South Carolina, Charleston, SC 29425, USA; walterho@musc.edu (S.W.); fitzgiwr@musc.edu (W.F.); 3Glaucoma Research Chair, Department of Ophthalmology, College of Medicine, King Saud University, Riyadh 11451, Saudi Arabia; akondkar@gmail.com or; 4Department of Pharmaceutical Sciences, Taneja College of Pharmacy, University of South Florida, Tampa, FL 33612, USA; biswal@usf.edu; 5Department of Ophthalmology, Medical University of South Carolina, Charleston, SC 29425, USA

**Keywords:** retinol-binding protein 4 receptor 2, RBPR2, RBP4, STRA6, all-*trans* retinol, photoreceptors, visual function, vitamin A

## Abstract

**Simple Summary:**

This work represents an initial evaluation of the second RBP4-vitamin A receptor RBPR2 in a mammalian model. We provide evidence that the membrane localized RBPR2 protein, under variable conditions of dietary vitamin A intake, plays an important role for dietary vitamin A transport to the eye for ocular retinoid homeostasis and visual function. These findings are of general interest, as disturbances in blood and ocular vitamin A homeostasis are linked to retinal degenerative diseases, which are blinding diseases. The animal model described here could also serve as an in vivo tool to study mechanisms related to retinal cell degeneration that are associated with vitamin A deficiency.

**Abstract:**

The systemic transport of dietary vitamin A/all-*trans* retinol bound to RBP4 into peripheral tissues for storage is an essential physiological process that continuously provides visual chromophore precursors to the retina under fasting conditions. This mechanism is critical for phototransduction, photoreceptor cell maintenance and survival, and in the support of visual function. While the membrane receptor STRA6 facilitates the blood transport of lipophilic vitamin A into the eye, it is not expressed in most peripheral organs, which are proposed to express a second membrane receptor for the uptake of vitamin A from circulating RBP4. The discovery of a novel vitamin A receptor, RBPR2, which is expressed in the liver and intestine, but not in the eye, alluded to this long-sort non-ocular membrane receptor for systemic RBP4-ROL uptake and transport. We have previously shown in zebrafish that the retinol-binding protein receptor 2 (Rbpr2) plays an important role in the transport of yolk vitamin A to the eye. Mutant *rbpr2* zebrafish lines manifested in decreased ocular retinoid concentrations and retinal phenotypes. To investigate a physiological role for the second vitamin A receptor, RBPR2, in mammals and to analyze the metabolic basis of systemic vitamin A transport for retinoid homeostasis, we established a whole-body *Rbpr2* knockout mouse (*Rbpr2^−/−^*) model. These mice were viable on both vitamin A-sufficient and -deficient diets. *Rbpr2^−/−^* mice that were fed a vitamin A-sufficient diet displayed lower ocular retinoid levels, decreased opsins, and manifested in decrease visual function, as measured by electroretinography. Interestingly, when *Rbpr2^−/−^* mice were fed a vitamin A-deficient diet, they additionally showed shorter photoreceptor outer segment phenotypes, altogether manifesting in a significant loss of visual function. Thus, under conditions replicating vitamin A sufficiency and deficiency, our analyses revealed that RBPR2-mediated systemic vitamin A transport is a regulated process that is important for vitamin A delivery to the eye when RBP4-bound ROL is the only transport pathway in the fasting condition or under vitamin A deficiency conditions.

## 1. Introduction

In humans, prolonged dietary vitamin A/all-*trans* retinol (ROL) deficiency (VAD) or inherited mutations in retinoid/visual cycle genes can lead to either ocular vitamin A deficiency or accumulation of toxic vitamin A byproducts, which can manifest in similar retinal degenerative phenotypes that can result in the loss of visual function and blindness [1,2,3,4,5,6,7,8,9,10,11,12,13,14,15,16,17,18,19,20,21,22,23,24,25,26,27,28,29,30,31,32,33,34,35,36]. Thus, an understanding of mechanisms that facilitate and regulate the uptake, transport, and long-term storage of dietary vitamin A for systemic retinoid homeostasis is significant to the design of strategies aimed at attenuating retinal degenerative diseases associated with ocular ROL deficiency or excess [2,5,13,37,38,39,40,41,42,43,44,45,46,47,48,49]. Dietary ROL bound to the retinol-binding protein 4 (RBP4-ROL) as a method of transport has been suggested to specifically function in the ocular vitamin A-dependent processes of vision in humans [5,6,7,8,9,10,11,12,13,14,15,16,17]. However, the precise mechanism(s) that facilitate and regulate this process for vision is not fully understood, and elucidation will be highly significant given the role vitamin A metabolites (retinoids) play in retinal health and disease states [2,3,4,5,6,7,8,17,19]. 

Biochemical and genetic studies from the Sun and von Lintig laboratories have elegantly shown, both in vitro and in vivo, the involvement of STRA6 (stimulated by retinoic acid 6) in the uptake of circulatory RBP4-ROL to the vertebrate eye [5,10,11,12]. STRA6 is not however expressed in the vertebrate (mouse and zebrafish) intestine, liver, and most peripheral vitamin A storage tissues proposed to express a second RBP4-ROL receptor [5,10,11,12,13]. Additionally, since circulating RBP4-ROL may be absorbed by extra-hepatic tissues or recycled back to the liver multiple times before it can be fully metabolized or degraded, this suggests the existence of a second RBP4-ROL receptor [5,13,33,49]. Furthermore, while we and others have implicated the scavenger receptor class B type I (SR-B1) receptor in the cellular uptake of dietary pro-vitamin A carotenoids for ROL production, SR-B1 is not involved in dietary ROL absorption [5,8,40,41,42,43]. Therefore, the systemic uptake and peripheral tissue storage of RBP4-ROL might be mediated by the retinol-binding protein 4 receptor 2 (RBPR2, also termed STRA6like) [5,11,13]. The RBPR2 protein was first identified by the Graham laboratory in 2013, where they biochemically implicated this non-ocular expressed membrane receptor in RBP4 binding and ROL transport [13]. 

We characterized the physiological function of Rbpr2 for photoreceptor health and visual function in zebrafish, where we elucidated the consequences of Rbpr2 deficiency for maternal yolk ROL mobilization to the developing eye using retinal cell development and visual function tests as sensitive end-point readouts [14,16]. Rpbr2 is a high affinity Rbp4 membrane receptor that facilitates long-term vitamin A storage in liver, intestine, and adipose, among other tissues [13,14,16]. Supporting this observation, Rbpr2 expression in zebrafish, similar to mice, was specific to the developing liver, intestinal enterocytes, and pancreas, the very tissues proposed to mediate systemic vitamin A uptake and participate in long-term ROL storage for vision [13,14,16]. In addition, since zebrafish mobilize maternal yolk vitamin A (>90%) [50,51,52] during development to aid in vision, consequently, in *rbpr2*-mutant zebrafish lines, we observed reduced ocular retinoid concentrations, shorter photoreceptor outer segments, and impaired ocular retinoid signaling, altogether manifesting in a loss of visual function [14,16]. The significance of our findings is that it was shown that the Rbpr2 receptor could physiologically function as the systemic ROL receptor for retinoid homeostasis and for vision in a vertebrate model [14,16]. Recently, RBPR2 has been implicated in the liver re-uptake of circulatory vitamin A in mutant *Stra6^−/−^*- and *Isx^−/−^*-deficient mice, further strengthening the proposed hypothesis of its in vivo involvement in systemic retinoid homeostasis [33].

Here we report the generation of whole-body *Rbpr2*-knockout mice to test the hypothesis that RBPR2 participates in dietary vitamin A transport and distribution to the eye for vision. Significantly, such membrane receptors could provide cells with a mechanism to regulate dietary retinol uptake and availability for vision [1,2,3,4,5,6,7,28,33,39,53,54,55,56,57,58]. These results extended the original insight into Rbpr2 function as a systemic Rbp4 receptor for retinoid homeostasis and for vision [13,14,16]. Together with previous reports, our findings identifying RBPR2 as a systemic RBP4-vitamin A receptor have important implications for disease states associated with impaired blood vitamin A homeostasis with relevance to chromophore production and visual function [13,33].

## 2. Materials and Methods

### 2.1. Materials

All chemicals, unless stated otherwise, were purchased from Sigma-Aldrich (St. Louis, MO, USA) and were of molecular or cell culture grade quality.

### 2.2. Generation of Rbpr2 Knockout Mice

We obtained cryopreserved conditional *Rbpr2* (*Stra6like*) embryos from the UC Davis KOMP Repository and generated *Rbpr2*floxed/wild-type (*Rbpr2^fl/wt^*) mice (Case Western Transgenic Core, Cleveland, OH, USA). Mice were generated on a C57B6/**6J** background to avoid potential problems with *Rd8* mutation (found in C57B6/**6N** lines) [59]. *Rbpr2^fl/wt^* pups were genotyped using genomic DNA from tail clips and KOMP-PCR primers CSD-loxF with CSD-R to identify the floxed allele mice. Since several reports have indicated that a *neo*^R^ selection cassette insertion into a target gene can result in nonspecific phenotypes, e.g., by influencing the expression of neighboring genes [12], heterozygous (*Rbpr2^fl/wt^*) male and female mice were therefore bred with the *Rosa^flp/flp^* mice (JAX labs) to finally obtain the *Rbpr2^fl/fl^*; *Rosa^flp/flp^* mice (annotated as *Rbpr2^fl/fl^* in this proposal). By crossing chimeras to a *FLP* deleter strain, the FRT-flanked *neo*^R^ selection marker was excised, and the conditional *Rbpr2*-loxP-flanked allele (neo^R^-free) was generated. Breeding pairs were fed purified rodent diets (AIN-93G; Research Diets, New Brunswick, NJ, USA) containing 8 IU of vitamin A/g.

### 2.3. Generation of Rbpr2 Knockout (Rbpr2^fl/fl^; Actin-Cre+) Mice

Previously, to examine the role of STRA6 for circulatory RBP4-ROL uptake into the eye, four groups generated *Stra6^−/−^*-deficient animals using either a conventional or Cre+ (whole-body) approach to disrupt STRA6 function [12,15,60,61]. In a similar strategy, to eliminate the systemic function of *Rbpr2* for dietary vitamin A transport^13^, we crossed *Rbpr2^fl/fl^* mice with Actin-Cre+ mice (JAX labs) [62] and obtained heterozygous *Rbpr2^fl/wt^*; Actin-Cre+ mice. Heterozygous *Rbpr2**^fl/wt^*; Actin-Cre+ mice were further mated with *Rbpr2**^fl/fl^* mice, and litters were genotyped to obtain the experimental *Rbpr2^fl/fl^*; Actin-Cre+ and littermate controls. *Rbpr2^fl/fl^*; Actin-Cre+ mice were then backcrossed with wild-type (WT) C57B6/**6J** mice for five generations. In this manuscript, the resulting experimental *Rbpr2^fl/fl^*; Actin-Cre+ mice are referred to as *Rbpr2^−/−^* mice. *Rbpr2^−/−^* adults, such as *Rbp4^−/−^* and *Stra6^−/−^* mice, were viable and fertile [7,12,15]. Breeding pairs (*Rbpr2^fl/fl^ and Rbpr2^fl/wt^*; Actin-Cre+) and their litters were kept with ad libitum access to food and water at 24 °C in a 12:12 h light–dark cycle. All mice experiments were approved by the Institutional Animal Care and Use Committee (IACUC protocol #00780) of the Medical University of South Carolina, SC, and (IACUC protocol #38982A) of the University of Minnesota, MN, USA, and performed in compliance with the ARVO Statement for the use of Animals in Ophthalmic and Vision Research. Equal numbers of male and female mice (50:50 ratio) were used per group and time point. Specially formulated and purified low vitamin A diets/vitamin A-deficient (VAD) diets contained 0.22 IU vitamin A/g based on the AIN-93G diet (Research Diets, New Brunswick, NJ, USA).

### 2.4. Genotyping of the Crb1 and Pde6b Locus

Whole genomic DNA samples were isolated from tail-biopsies of breeding pairs and WT mice and amplified separately for the WT allele and mutant *Rd8* and *Rd1* alleles, as described previously [59,62,63,64]. All animals used in this study were confirmed to be negative for the known *Rd8* and *Rd1* mutations.

### 2.5. Western Blot Analysis to Detect RBPR2 Expression in Mice

To confirm RBPR2 loss in *Rbpr2^−/−^* mice, two custom rabbit polyclonal antibodies towards mouse RBPR2 were generated (FabGennix, Frisco, TX, USA). These RBPR2 antibodies are predicted not to cross-react with mouse STRA6 (Appendix A). Western blot analysis of tissues from *Rbpr2^−/−^* mice showed that RBPR2 protein expression was undetectable in these animals compared to WT mice; thus, we confirmed that RBPR2 was genetically disrupted in systemic tissues of *Rbpr2^−/−^* mice. RBPR2 peptide information is provided in Appendix A.

### 2.6. Semi-Quantitative PCR Analysis for Rbpr2 and Stra6 mRNA Expression in Mice

Total RNA isolation from WT mouse tissues (*n* = 3 mice) was carried out using TRIzol reagent (Invitrogen, Carlsbad, CA, USA) according to previously published methods [41,42]. Samples were pooled, and RNA concentration and purity were measured using a NanoDrop spectrophotometer (ND-100, ThermoScientific, Waltham, MA, USA). An Applied BioSystems reverse-transcription kit (Applied Biosystems, Waltham, MA, USA) was used to reverse transcribe 1.0 µg of RNA to cDNA. Semi-quantitative PCR was performed using gene-specific *Rbpr2* and *Stra6* primers. A β-Actin gene-specific primer pair was used as an endogenous control.

### 2.7. Immunohistochemistry and Fluorescence Imaging

Light-adapted mice were euthanized and their eyes enucleated. Eyes were then fixed in 4% paraformaldehyde buffered with 1X PBS for 2 h at 4 °C using established protocols [63,64,65]. Primary antibodies used in this study were diluted in blocking solution as follows: anti-rhodopsin/1D4 (1:500, Abcam, Waltham, MA, USA), anti-red/green cone opsin (M-opsin; 1:500; Millipore, St. Louis, MO, USA), and 4′,6-diamidino-2-phenylendole (DAPI; 1:5000, Invitrogen, Waltham, MA, USA) or Hoechst (1:10,000, Invitrogen) was used to label nuclei. All secondary antibodies (Alexa 488 or Alexa 594) were used at 1:5000 concentrations (Molecular Probes, Eugene, OR, USA). Optical sections were obtained with a Leica SP8 confocal microscope (Leica, Wetzlar, Germany) and processed with Leica Viewer software [66,67]. All fluorescently labeled retinal sections on slides were analyzed using BioQuant NOVA Prime Software (R & M Biometrics, Nashville, TN, USA), and fluorescence within individual retinal layers was quantified using *Image J* or Fiji (imagej.nih.gov, accessed on 5 January 2022). Briefly, immunostained retinal images were exported and opened using the *Image J* software, and the free hand tool was used to draw the region of interest (ROI) in the retinal layers. From the Analyze menu, measurements were set to calculate area-integrated intensity and mean gray value. The background values were noted in areas with no fluorescence. Corrected fluorescence was measured as difference of integrated density and area, adjusted for background fluorescence. The quantified values from the multiple ROIs were plotted and then averaged. Statistical significance was measured using a two-tailed *t*-test (Gaussian distribution, unpaired). Significance was considered as having a *p*-value <0.05. Statistical analysis was done using GraphPad Prism.

### 2.8. Measurement of Photoreceptor Outer Nuclear (ONL) Thickness and Outer Segment (OS) Lengths of Mice Retinas

The lengths of the photoreceptor OS in WT, *Rbpr2^+/^*^−^, and *Rbpr2^−/−^* animals (from H&E sections of retinas) were imaged (Keyence BZ-X800 microscope) and measured at 12 consecutive points (at 150 μm distances) from the optic nerve (ON), as previously described by us [64]. Data from this analysis were combined and are represented as total photoreceptor length (PRL) and ONL thickness (Area µm^2^). Retinal sections (*n*= 5–7 retinal sections per eye) from *n* = 8 mice for each genotype and time-point were analyzed, as previously described [64]. Two-way ANOVA with Bonferroni post-tests compared *Rbpr2^−/−^* to WT mice at each segment measured.

### 2.9. Electroretinogram (ERG) Analysis

Dark-adapted WT and *Rbpr2^−/−^* mice (50:50 ratio of male and female; *n* = 8 each genotype) were anesthetized by intraperitoneal injection of a ketamine/xylene anesthetic cocktail (100 mg/kg and 20 mg/kg, respectively), and their pupils were dilated with 1% tropicamide and 2.5% phenylephrine HCl. ERGs were performed under dim red light in the ERG rooms in the morning (8 am–11 am). Scotopic ERGs were recorded with a computerized system (UTASE-3000; LKC Technologies, Inc., Gaithersburg, MD, USA), as previously described [63,64,65].

### 2.10. Transmission Electron Microscopy (TEM) Analysis of Retinas

At the indicated time-points, eyecups were harvested and fixed overnight at 4 °C in a solution containing 2% paraformaldehyde/2.5% glutaraldehyde (buffered in 0.1 M cacodylate buffer), as previously described by us [63,64]. For TEM analysis, each eye (*n* = 6 individual eyes from *n* = 6 animals of each genotype) was cut in half before embedding in Epon blocks. Images were acquired with a Hamamatsu camera and software. All samples were processed by the Electron Microscopy Resource Laboratory at the Medical University of South Carolina, as previously described [63,64,65].

### 2.11. High-Performance Liquid Chromatography (HPLC) Analyses of Retinoids

Retinoid isolation procedures were performed under a dim red safety light (600 nm) in a dark room. Briefly, 2M hydroxylamine (200 μL) was added to the eyecup before the tissue was homogenized. Retinoid extraction was performed twice with a mixture containing 200 μL of methanol, 400 μL of acetone, and 500 µL of hexane. HPLC analysis was performed on a normal-phase Zorbax Sil (5 μm, 4.6 × 150 mm) column (Agilent, Santa Clara, CA, USA). Chromatographic separation was achieved by isocratic flow of 10% ethyl acetate/90% hexane at a flow rate of 1.4 mL/min for 35 min. Retinols and retinyl esters were detected at 325 nm using a UV–Vis DAD detector. For quantifying molar amounts of retinoids, the HPLC was previously scaled with synthesized standard compounds as previously described by us [41,42,43]. Calculation of concentration (µM): Standards were injected in concentrations ranging from 0 to 3.5 µM prepared solutions in mobile phase. The plotted concentrations were fit through linear regression to obtain the R-equation (y = mx + c) where y is the peak area (mAU*sec), m is the slope of the calibration curve, and c is the y-intercept. The area from the HPLC peaks of the samples (mAU*sec) are interpolated into concentration and expressed as picomoles. For eyes, the values are expressed as picomoles/eye; for liver, the values are expressed as picomoles/mg; for serum, the values are expressed as picomoles/microliter.
$$Concentration\;X\;\left(picomoles\right)=\frac{Peak\;Area\;Y\;\left(mAU*Sec\right)+Y-intercept}{Slope\;\left(m\right)}$$

### 2.12. Spectral-Domain Optical Coherence Tomography (SD-OCT) Analysis

Pupils were fully dilated with a 1% tropicamide solution (Falcon Pharmaceuticals, Fort Worth, TX, USA), and mice were anesthetized with an intraperitoneal injection of ketamine (100 mg/kg, Ketaset, Fort Dodge, IA, USA) and xylazine (4 mg/kg, AnaSed, Lloyd Laboratories, Shenandoah, IA, USA). Whiskers were trimmed to avoid image artifacts. OCT images were acquired with linear B-scan mode by employing ultra-high-resolution SD-OCT (Bioptigen).

### 2.13. Fundus Acquisition and Analysis

Mouse fundus imaging was performed with a cSLO (SpectralisHRA2, Heidelberg Engineering, Heidelberg, Germany) with a 55° lens. The near-infrared reflectance image (IR mode, 820 nm laser) was used to align the fundus camera relative to the pupil to obtain an evenly illuminated fundus image. An ICGA mode (790 nm) laser was used for angiography 10 min after mice were injected intraperitoneally with ICG (15 mg/kg, Acros Organics, NJ, USA).

### 2.14. Western Blot Analysis and Densitometry for Protein Quantification

Total protein from mouse retinal tissue (*n* = 4 per genotype) was extracted using the M-PER protein lysis buffer (ThermoScientific, Beverly, MA, USA) containing protease inhibitors (Roche, Indianapolis, IN, USA). Approximately 25 μg of total protein was electrophoresed on 4–12% SDS-PAGE gels and transferred to PVDF membranes. Membranes were probed with primary antibodies against CRALBP1 (1:100, Invitrogen), rod transducin/GNAT1 (1:250, Santa Cruz), PKC (1:500, Novus Biologicals), and β-actin or Gapdh (1:10,000, Sigma) in antibody buffer (0.2% Triton X-100, 2% BSA, 1X PBS). HRP conjugated secondary antibodies (BioRad, Hercules, CA, USA) were used at 1:10,000 dilution. Protein expression was detected using an LI-COR Odyessy system, and relative intensities of each band were quantified (densitometry) using *Image J* software version 1.49 and normalized to their respective loading controls. Each Western blot analysis was repeated thrice.

### 2.15. Statistical Analysis

Data were expressed as means ± standard deviation by ANOVA in Statistica 12 software (StatSoft Inc., Tulsa, Oklahoma, USA). Differences between means were assessed by Tukey’s honestly significant difference (HSD) test; *p*-values below 0.05 (*p* < 0.05) were considered statistically significant. For Western blot analysis, relative intensities of each band were quantified (densitometry) using *Image J* software version 1.49 and normalized to the loading control β-actin. Statistical analysis was carried out using PRISM 8 software-GraphPad. 

## 3. Results

### 3.1. Confirming the Loss of RBPR2 Expression in Whole Body Rbpr2-Knockout (Rbpr2^fl/fl^; Actin Cre+) Mice

To test the physiological role of RBPR2 for systemic dietary ROL uptake and transport for ocular retinoid homeostasis and visual function, we generated a whole-body *Rbpr2*-knockout mouse (Materials and Methods; Figure 1A). The *Rbpr2^fl/fl^*; Actin-Cre+ mice are referred to as *Rbpr2^−/−^* mice in this manuscript and were born in Mendelian ratio (Figure 1B). *Rbpr2^−/−^* adults, such as previously generated *Rbp4^−/−^* and *Stra6^−/−^* mice, were viable and fertile [7,12,15]. To confirm RBPR2 loss in *Rbpr2^−/−^* mice, a custom rabbit polyclonal antibody was generated, and a Western blot analysis of various tissues from *Rbpr2^−/−^* mice confirmed that RBPR2 protein expression was undetectable in these animals, compared to wild-type (WT) mice (Figure 1C and Appendix A). Hence, we concluded that RBPR2 was genetically disrupted in the systemic tissues of these mice, and we designated these mice as *Rbpr2* knockout (*Rbpr2^−/−^*) mice [13,15]. These mice and their heterozygous siblings (*Rbpr2^+/−^*) or *Rbpr2^fl/fl^* mice, together with age-matched WT mice, were used in the experiments described below.

### 3.2. Rbpr2 and Stra6 mRNA Expression in Mice Tissues

Semi-quantitative RT-PCR analysis of murine tissues in adult WT animals showed that Rbpr2 mRNA expression was highest in the liver and kidneys, followed by a lower expression in extra-ocular tissues, including the intestines (Figure 2A). These are the very organs known to be involved in dietary vitamin A uptake and/or long-term storage [5,49]. As reported previously in WT-FVB mice, no Rbpr2 mRNA expression was detected in the eyes of WT-C57/B6 animals (Figure 2A,B) [13]. Conversely, and as reported previously, we observed that Stra6 mRNA expression in WT mice was highest in the eyes and kidneys, but undetectable in the liver [12,13,15,33].

### 3.3. Rbpr2^−/−^ Mice on Vitamin A-Sufficient Diets Show Decreased Ocular Retinoid Concentrations and Reduced Visual Function

To investigate a physiological role for the second RBP4-vitamin A receptor, RBPR2, in mammals and to analyze the metabolic basis of systemic vitamin A transport for ocular retinoid homeostasis, we fed 4-week old WT, *Rbpr2^fl/fl^, Rbpr2^+/−^*, and *Rbpr2^−/−^* mice (*n* = 8 each genotype; 50:50 ratio male and female) on vitamin A-sufficient (VAS) diets (8 IU/g) for 8-weeks, after which the major vitamin A/ROL storage organs, including the eyes, were harvested (mice were 12 weeks old at the time of analysis). Histological analysis revealed that systemic organs (liver, heart, and kidney shown) of WT, *Rbpr2^+/−^*, and *Rbpr2^−/−^* mice (3-months of age) had unremarkable pathology, indicating that retinoid signaling in these organs was likely maintained by circulatory retinyl esters (RE) in chylomicrons (Appendix A); however, this hypothesis requires further investigation and more detailed analysis in this mutant animal model. To test if loss of RBPR2 affected ocular retinoid concentrations, we performed HPLC analysis for retinoids in the eyes of controls and *Rbpr2^−/−^* mice. This analysis showed that in contrast to WT and heterozygous *Rbpr2^+/−^* mice, retinas of *Rbpr2^−/−^* mice showed a significant decrease in ocular retinoid concentration (~35–51% decrease; * *p* < 0.05) compared to controls (Figure 3A and Appendix A). Additionally, these mice also showed decreased liver and serum ROL concentrations (** *p* < 0.005; Appendix A). Accordingly, scotopic ERG responses (both *a*- and *b*-wave) were significantly reduced (* *p* < 0.05) in the eyes of *Rbpr2^−/−^* mice when compared to controls at 12 weeks of age (Figure 3B,C).

However, while we did observe a modest shortening of the photoreceptor outer segments (OS) in the retinas of young *Rbpr2^−/−^* mice fed a VAS diet, this did not reach statistical significance (Appendix A). Interestingly, we observed depigmentation (fewer melanosomes) of the retinal pigmented epithelium (RPE) of *Rbpr2^−/−^* mice, which was similar to what we observed previously in the RPE of *rbpr2^−/−^* mutant zebrafish [14,16]. We next analyzed the morphology of living eyes of 12 weeks old control and *Rbpr2^−/−^* mice by OCT. This analysis revealed an overall intact lamination of retinal layers in *Rbpr2^−/−^* mice compared to controls (Appendix A; ONL thickness and PR layer length quantified in Appendix A). Angiography after an indocyanine green (ICG) injection further revealed no pathological vessel leakages in the *Rbpr2^−/−^* mice when compared to controls (Appendix A). Thus, young *Rbpr2^−/−^* mice showed decreased ocular retinoid concentrations and reduced visual function when fed vitamin A-sufficient diets.

### 3.4. Rbpr2^−/−^ Mice on Vitamin A-Deficient Diets Show Photoreceptor Phenotypes

Previous studies have indicated that the vitamin A status of most tissues can be maintained by alternative mechanisms that involve retinyl esters (RE) in chylomicrons when copious amounts of dietary vitamin A are available [5,49]. Hence, we next tested whether this picture could change when these animals are subjected to prolonged dietary vitamin A deficiency (VAD). Therefore, to test the dependence of visual function on RBPR2 in the fasting state (i.e., when humans are not actively consuming vitamin A-rich foods), we fed 4-week old WT, *Rbpr2^fl/fl^*, littermate heterozygous *Rbpr2^+/−^*, and homozygous *Rbpr2^−/−^* mice (*n* = 8 each genotype; 50:50 ratio of males and females) low vitamin A diets/vitamin A-deficient (VAD) diets (0.22 IU/g) for 8-weeks and then harvested the eyes (mice were 12 weeks old at the time of analysis). Histological analysis of systemic organs from 12-week old WT, *Rbpr2^+/−^*, and *Rbpr2^−/−^* mice had unremarkable pathology, indicating that retinoid signaling in these peripheral organs was likely maintained by circulatory RE in chylomicrons (Figure 4A and Appendix A), an observation and hypothesis that requires more detailed analysis.

Interestingly, *Rbpr2^−/−^* mice on vitamin A-deficient diets showed photoreceptor anomalies, previously reported in *Stra6^−/−^* mice and in vitamin A-deficient retinas [7,12,17]. Histological analysis showed that *Rbpr2^−/−^* mice on VAD diets had significantly thinner photoreceptor OS cell layers (Figure 4B, OS lengths quantified in Figure 4C; ** *p* < 0.005). The TEM study confirmed that outer segments (OS) in *Rbpr2^−/−^* mice were ~36–54% shorter (*p* < 0.005) when compared to controls (Figure 5A). HPLC analysis further revealed an even more significant decrease in total retinoid concentration (~75–80% decrease; ** *p* < 0.005) in the eyes of *Rbpr2^−/−^* mice that were fed vitamin A-deficient diets compared to controls or *Rbpr2^−/−^* mice on vitamin A-sufficient diets (Figure 5B vs. Figure 3A; Appendix A vs. Appendix A).

### 3.5. Rbpr2^−/−^ Mice on Vitamin A-Deficient Diets Show Severely Reduced Visual Responses

Using OCT analysis, we observed that the photoreceptor (PRL) and outer nuclear layer (ONL) in *Rbpr2^−/−^* mice fed VAD diets was significantly thinner than those observed in age-matched WT mice (** *p* < 0.005 for photoreceptor cell layer; Figure 6A and Appendix A). Since atrophy of the choroid and choroidal neovascularization has been previously observed in *Stra6**^−/−^*-deficient mice and in patients with RBP4 mutations [12,68,69,70], we wished to investigate this phenotype in living eyes of *Rbpr2**^−/−^* mice. Angiography after an indocyanine green (ICG) injection revealed pathological choroidal vessel leakages in the *Rbpr2^−/−^* mice fed VAD diets, in contrast to controls (*p* < 0.05; Figure 6B). Accordingly, scotopic ERG responses were significantly diminished in the eyes of *Rbpr2^−/−^* mice on VAD diets when compared to age-matched controls at 12 weeks of age (** *p* < 0.005, for both *a*- and *b*-waves, *Rbpr2^−/−^* vs. controls; Figure 6C,E vs. Figure 6D,F).

### 3.6. Rod and Cone Opsin Expression Are Reduced in Rbpr2^−/−^ Mice

To test the effects of decreased ocular retinoids on rod and cone opsin expression, we stained retinal sections of 12-week old control and *Rbpr2^−/−^* mice (on VAS and VAD diets) using rhodopsin and R/G cone opsin antibodies. Control and *Rbpr2^−/−^* mice on both VAS and VAD diets showed loss of rhodopsin expression (*p* < 0.05), mislocalization of rhodopsin, and decreased cone opsin staining (*p* < 0.05). Additionally, *Rbpr2^−/−^* mice on VAD diets showed shorter OS when compared to controls and as previously observed in vitamin A-deficient retinas [12] (Figure 7A,B vs. Figure 7C,D; rhodopsin quantified in Figure 7E and cone M-opsin quantified in Figure 7F). We then analyzed the expression patterns of G protein (GNAT1), a rod-specific component of the phototransduction cascade, cellular retinaldehyde-binding protein 1 (CRALBP1), and protein kinase C (PKC), which is involved in the retinal vasculature in *Rbpr2^−/−^* mice on VAS diets. While the expression of CRALBP1 was unaffected in *Rbpr2^−/−^* mice, however, we observed a decrease in PKC (*p* < 0.05) and GNAT1 expression (*p* < 0.05) in these mice compared to WT controls and when normalized to the protein loading control (Figure 7G; quantified in Figure 7H).

## 4. Discussion

Dietary vitamin A (all-*trans* retinol/ROL) must be adequately absorbed, stored, and distributed within the mammalian body to produce visual chromophores in the eyes and all-*trans* retinoic acid (a*t*RA) in tissues. All-*trans* ROL is the major transport form of dietary vitamin A and is transported via blood bound to the RBP4 protein as RBP4-ROL (holo-RBP4) (Figure 8). Previously, it has been suggested that a receptor-mediated mechanism is required to facilitate circulatory ROL-bound RBP4 uptake, storage, and transport in systemic tissues, which is important for vision in humans [1,2,3,4,5,6,7,8,9,10,11,12,13,17,49,53,54,55]. While it is well established that the RPE cells in the retina express a membrane-bound receptor, STRA6, which binds to holo-RBP4 and facilitates the internalization of all-*trans* ROL into the eye, however, non-ocular tissues such as the intestine, which is known to absorb and efflux vitamin A, or the liver, which is the major storage organ for dietary vitamin A, do not express STRA6 [5,10,11,12,13,15]. Based on the 2013 publication from the Graham laboratory, where they identified a novel membrane receptor in the liver, RBPR2, for RBP4-ROL transport, we further reasoned that this second vitamin A receptor, RBPR2, which is highly expressed in non-ocular tissues in mammals and previously shown in zebrafish to be essential for yolk vitamin A transport to the eye, may play a physiological role in systemic ROL-bound RBP4 uptake and transport, in maintaining whole-body retinoid homeostasis, and in support of visual function (Figure 8).

Here, we established a global *Rbpr2*-knockout mouse (*Rbpr2^−/−^*) and investigated the consequences of systemic RBPR2 deficiency in dietary ROL transport. We show here that RBPR2, except for the eye, is widely expressed in systemic or non-ocular tissues, including the intestine and liver, the very organs known to transport dietary ROL or be involved in its long-term storage, respectively. Under variable conditions of dietary vitamin A in the diet, *Rbpr2^−/−^* mice show depleted ocular retinoid concentrations and decreased visual function. Significantly, we found that the systemic RBP4-ROL/RBPR2 transport system is critical for ocular retinoid homeostasis, especially under fasting conditions, when RBP4-ROL is the only means of vitamin A supply to the eye. Herein, when *Rbpr2^−/−^* mice were challenged with a vitamin A-deficient diet, these mice showed significantly depleted ocular retinoid concentrations, decreased levels of opsin proteins, and shorter photoreceptor OS phenotypes that manifested in a loss of visual function (Figure 8). Thus, our work, therefore, establishes in a mammalian model a physiological role for RBPR2 in the systemic transport of ROL to the eye in support of retinal homeostasis and visual function.

## 5. Differential Gene Expression Patterns of RBPR2 and STRA6 in Vertebrates

First, using a semi-quantitative RT-PCR approach, we investigated the tissue expression patterns of STRA6 and RBPR2 in wild-type adult mice. As reported previously, we confirmed that while STRA6 was highly expressed in the eyes and kidneys, it was not expressed in the intestine and liver. Conversely, RBPR2 mRNA expression was robustly found to be expressed in the murine intestine, liver, spleen, and non-ocular tissues, but not in the eyes. Since the liver displays a high affinity for RBP4-ROL uptake and ROL efflux, it has been postulated that specific membrane receptors that facilitate vitamin A transport, other than STRA6, might be present [5,8,13,49]. Based on our previous observations in zebrafish [14,16,71] and those of others in zebrafish and mice [10,12,13,15], it is likely that the second vitamin A receptor, RBPR2, identified by the Graham group in 2013, functions as the systemic receptor for vitamin A in non-ocular tissues, while STRA6 functions as the exclusive vitamin A receptor in the eye [5,49]. Interestingly, both RBPR2 and STRA6 were co-expressed in the kidneys, where they might coordinate a yet unknown function in renal homeostasis or play a role in the retention of dietary retinol [5,49,72,73,74]. In a recent study, from the von Lintig group, in *Stra6^−/−^* and *Isx^−/−^*-deficient mice, a greater than 100-fold higher *Rbpr2* mRNA expression and increased liver retinoid concentrations was observed in these mutant mice [33]. This indicates that the systemic retinol receptor, RBPR2, is physiologically involved in the reuptake of RBP4-ROL from the circulation, to prevent toxic accumulation of excessive amounts of ROL in these mutant mice [33]. 

## 6. Loss of RBPR2 Influences Ocular Retinoid Homeostasis and Visual Function

The observation that the systemic organs of *Rbpr2**^−/−^* mice on vitamin A-sufficient diets had unremarkable histology indicated that retinoid signaling in these organs was likely maintained by circulatory retinyl esters in chylomicrons [5,49,74]. However, in contrast to WT mice, HPLC analysis of retinas of *Rbpr2**^−/−^* mice showed a significant decrease in ocular retinoid concentrations and a decrease in ERG amplitudes, but with no retinal pathology. Conversely, 12-week old *Rbpr2**^−/−^* mice that were fed vitamin A-deficient diets showed striking retinal phenotypes, where we observed loss of opsin staining/decreased levels of the chromophore, shorter photoreceptor OS, decreased ERG responses, indicating that photoreceptor health in *Rbpr2**^−/−^* mice, such as that in *Stra6^−/−^* mice^12^, are more susceptible to low dietary vitamin A intake (i.e., in the fasting state and when humans are not actively consuming vitamin A-rich foods). Cumulatively, these results suggest that (i) the ocular vitamin A status in *Rbpr2^−/−^* animals under either dietary vitamin A condition is extremely tenuous, particularly concerning dietary vitamin A uptake and/or transport for ocular retinoid homeostasis, (ii) RBPR2 is critical for maintaining ocular retinoid homeostasis and photoreceptor health, especially in the fasting state, and (iii) there is a more direct role for RBPR2 in the transport of dietary vitamin A for visual function^6,7^ (Figure 8).

## 7. Phenotypic Differences between *rbpr2^−/−^* Zebrafish and *Rbpr2^−/−^* Mice

The above observations in *Rbpr2**^−/−^* mice are in contrast to *rbpr2**^−/−^* mutant zebrafish, where we observed significant retinal pathology at early larval stage [14,16,71]. We can hypothesize on these differences. It is well-known that zebrafish develop within 5.5 days with well-established eyes and visual function. To support this process, embryos mobilize >90% of vitamin A stores from the egg yolk to the eye. Evidently, genetic targeting of vitamin A receptor, *rbpr2*, or s*tra6* in zebrafish provokes an early loss of yolk vitamin A transport to the eye, which results in a significant reduction of ocular retinoid concentration, manifesting in microphthalmia and retinal phenotypes at early larval stages [14,16,45,46,71]. Even though young *Rbpr2**^−/−^* mice on vitamin A-sufficient diets showed decreased ocular retinoid concentrations and reduced opsins, which resulted in decreased visual responses, it is possible that age or the concerted interplay between vitamin A receptors and LRAT may be a determining factor for the observation of retinal phenotypes, evident only when both systemic and ocular retinoid concentrations become severely depleted as these *Rbpr2**^−/−^* mice age [12,13,33,47]. We are currently investigating this hypothesis in aged *Rbpr2**^−/−^* mice from ages 6 to 18 months. 

## 8. Role of RBPR2 in Maintaining the Stoichiometry of Chromophore Production

An analysis of retinas/photoreceptors in *Rbpr2^−/−^* mice also indicated that RBPR2 is essential to maintain the stoichiometry of opsins and chromophore production. While the retinal lamination layers were generally intact in young *Rbpr2^−/−^* mice, conversely, photoreceptors showed decreased levels of M-cone opsins and rod opsins, together with mislocalized rod opsins in *Rbpr2^−/−^* mice on VAD diets. RBPR2 deficiency also affected the expression of rod-specific G proteins and PKC, essential components of the phototransduction cascade. This is similar to those observed in other genetic mouse models of vitamin A deficiency [75,76]. Thus, in the future it would be fascinating to study the opsin-chromophore imbalances (by additional analysis of cone S-opsins) in aged *Rbpr2**^−/−^* mice to understand its systemic role in balancing ocular retinoid levels and in adjusting it to chromophore levels throughout the life cycle. 

## 9. Study Limitations and Future Directions

These studies have limitations, as we did not study all possible biological functions of RBPR2 that may exist when compared to STRA6 [2,11,12,53,56]. For instance, the retinal phenotypes and decreases in ocular retinoid concentrations in the whole-body *Rbpr2^−/−^* mice could be related to either the loss of vitamin A efflux (in the intestine) or uptake and/or efflux (in the liver), where RBPR2 is known to be expressed [13] (Figure 2 and Figure 8). Additionally, decreased liver and serum ROL concentrations in *Rbpr2^−/−^* mice (Appendix A) suggest that either circulatory retinol uptake or efflux may be compromised in these mice. Therefore, to understand and investigate the possible tissue specific biological functions of RBPR2 independently and in concert with LRAT, we are currently generating intestine-specific (*Rbpr2^fl/fl^*;Villin-Cre+) and liver-specific (*Rbpr2^fl/fl^*; Albumin-Cre+) conditional KO mice. Additionally, the present results may simply support the fact that the eye has a very high demand for vitamin A and so when vitamin A delivery is compromised (i.e., in *Rbpr2^−/−^* mice), the greatest effect is on this tissue. However, there may be minor effects on other non-ocular tissues with lesser vitamin A demand that are not evident using the current microscopic approaches used in this study that require a more detailed analysis but is beyond the scope of the present study.

## 10. Conclusions

In summary, we observed in *Rbpr2^−/−^* mice that the eye takes up postprandial retinol very poorly compared to non-ocular tissues because systemic tissues other than the eye are capable of obtaining sufficient vitamin A; therefore, *Rbpr2^−/−^* mice are phenotypically normal except for their vision. As previously established in *Stra6* and *Rbp4* mice, in the fasting state the eye relies primarily on retinol bound to RBP4 (RBP4-ROL) as its main source for acquiring ocular retinoids needed for the production of the photopigments and normal vision. As *rd1* and *rd8* mutations were not observed in our mice, retinal function impairment could be caused by a deficiency in systemic retinol transport to the eye, which is likely dependent on the function of RBPR2. Thus, by understanding the mechanisms that control systemic and ocular retinoid homeostasis in sustaining chromophore synthesis, *Rbpr2^−/−^* mice will be a valuable future mammalian model to investigate these and other unresolved questions related to systemic vitamin A homeostasis and its relationship to eye physiology and in the treatment of blinding diseases associated with either hypovitaminosis/decreased serum ROL levels that could affect ocular retinoid concentrations, photopigment production, and photoreceptor health, or under conditions of vitamin A excess as in Stargardt disease, where excessive vitamin A transport to the eye can lead to accelerated production of the cytotoxic *bis*-retinoid A2E/lipofuscin [19,20,21].

## Figures and Tables

**Figure 1 nutrients-14-02371-f001:**
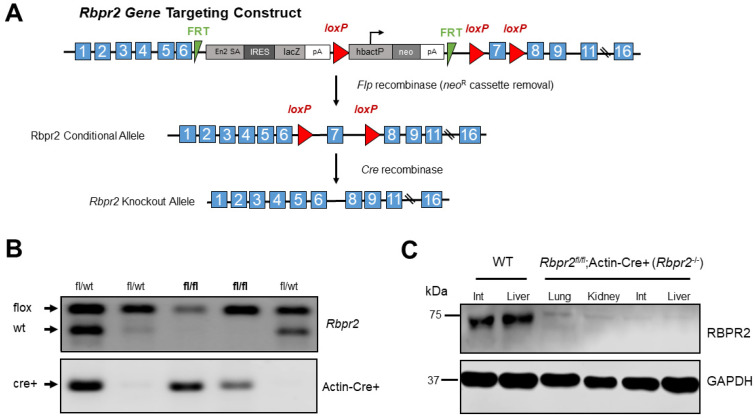
**Disruption of the mouse *Rbpr2* gene using the Cre/LoxP strategy and generation of the global *Rbpr2* knockout (*Rbpr2******^−/−^*****) mice.** (**A**) An *Rbpr2* targeting vector was designed to replace exon 7 with a neomycin cassette (*neo^R^*) flanked by *loxP* sites. This resulted in the *Rbpr2 neo^R^* cassette allele. The *neo^R^* cassette was floxed out by mating *Rbpr2^fl/wt^* mice with the *Rosa^flp/flp^* mice. The *Rbpr2^fl/fl^* mice were then mated with Actin-Cre+ mice to finally generate littermate controls and the global *Rbpr2* knockout (annotated as *Rbpr2**^−/−^*) mice. (**B**) PCR-based genotyping for identifying the *Rbpr2^fl/fl^*; *Actin-Cre+* (*Rbpr2**^−/−^*) and littermate control mice. (**C**) A custom made RBPR2 antibody was generated and used to determine the systemic loss of RBPR2 protein expression in *Rbpr2**^−/−^* mice. Immunoblot of protein extracts of tissues from adult WT and *Rbpr2**^−/−^* mice. Pooled protein from *n* = 3 mice (per genotype) at 6–8 weeks of age. GAPDH antibody was used as the loading control. Int, intestine; WT, wild-type.

**Figure 2 nutrients-14-02371-f002:**
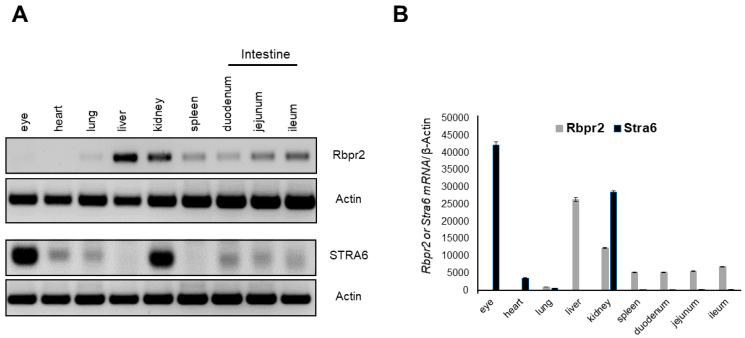
***Rbpr2* and *Stra6* mRNA expression in tissues of adult mice.** (**A**) Tissues were harvested from adult wild-type (WT) mice (*n* = 3). Total RNA was isolated and pooled together, respectively. Semi-quantitative RT-PCR analysis for *Rbpr2* and *Stra6* mRNA expression was performed using β-actin as the loading control and the respective gene-specific primers. (**B**) Numbers on the ordinate represent *Rbpr2* and *Stra6* mRNA expression levels normalized to mouse β-actin and expressed as mean ± SEM.

**Figure 3 nutrients-14-02371-f003:**
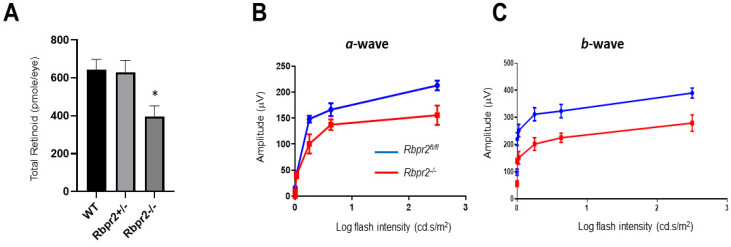
***Rbpr2**^−/−^* mice on vitamin A-sufficient diets show decreased ocular retinoid concentration and loss of visual function.** (**A**,**B**) Quantification of total retinoid concentrations (pmole/eye) in eyes from WT, *Rbpr2^+/−^*, and *Rbpr2**^−/−^* mice (pooled eyes from *n* = 8 mice per genotype). * *p* < 0.05, *Rbpr2**^−/−^* compared to controls. (**C**) Scotopic electroretinograms of 12-week old control (*Rbpr2^fl/fl^*) and *Rbpr2**^−/−^* mice on vitamin A-sufficient (VAS) diets. * *p* < 0.05, for both *a*- and *b*-wave; *Rbpr2^−/−^* mice vs. controls.

**Figure 4 nutrients-14-02371-f004:**
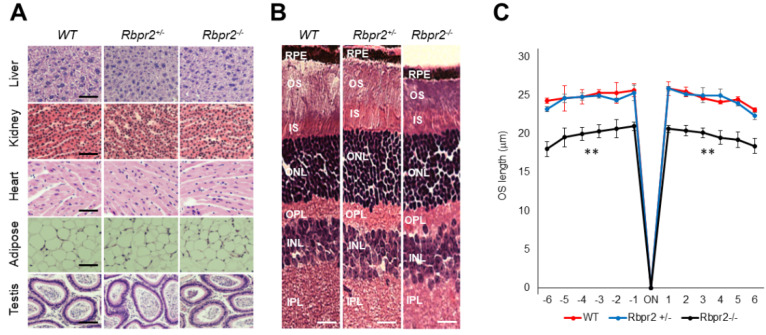
***Rbpr2**^−/−^* mice on vitamin A-deficient diets show retinal phenotypes.** (**A**,**B**) Representative tissue histology and staining of liver, heart, adipose, testis, retina (H&E stain), and kidney of 12-week old wild-type (WT), heterozygous *Rbpr2^+/−^*, and homozygous *Rbpr2**^−/−^* (KO) mice on vitamin A-deficient diets. (**C**) Quantification of photoreceptor outer segment (OS) lengths from H&E sections of 12-week old WT, heterozygous *Rbpr2^+/−^*, and homozygous *Rbpr2**^−/−^* (KO) mice using “spider graph” morphometry. The OS lengths from H&E sections through the optic nerve (ON; 0 μm distance from the optic nerve and starting point) were measured at 12 locations around the retina, six each in the superior and inferior hemispheres, each equally at approximately 150 μm distances. RPE, retinal pigmented epithelium; OS, outer segments; IS, inner segments; ONL, outer nuclear layer; INL, inner nuclear layer; IPL, inner plexiform layer. Scale bar = 50 µm (**A**); 100 µm (**B**). ** *p* < 0.005; *Rbpr2^−/−^* mice vs. controls.

**Figure 5 nutrients-14-02371-f005:**
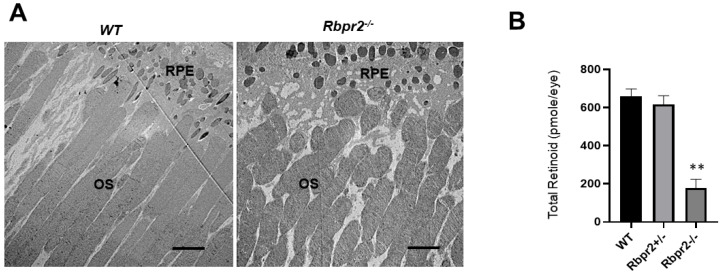
***Rbpr2**^−/−^* mice on vitamin A-deficient diets show decreased ocular retinoid concentrations and shorter photoreceptor outer segments.** (**A**) Ultrastructural analysis of photoreceptors using transmission electron microscopy (TEM): Representative TEM images of photoreceptors from 12-week old WT and *Rbpr2**^−/−^* mice on vitamin A-deficient diets are presented. Scale bar = 600 μm. RPE, retinal pigmented epithelium; OS, outer segments. Data are representative of *n* = 6 retinal sections per eye from *n* = 4 mice per genotype. (**B**) Quantification of total retinoid concentrations (pmole/eye) in eyes from WT, *Rbpr2^+/−^*, and *Rbpr2**^−/−^* mice (eyes from *n* = 8 mice per genotype). ** *p* < 0.005, *Rbpr2**^−/−^* compared to controls.

**Figure 6 nutrients-14-02371-f006:**
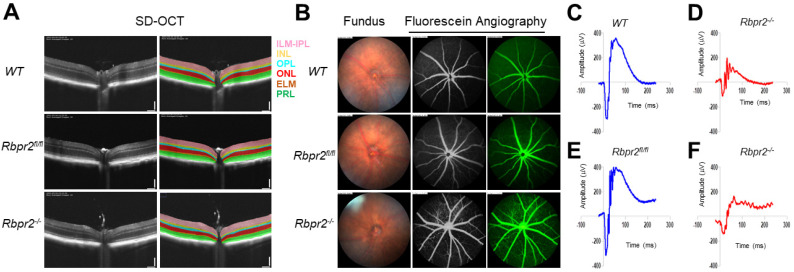
***Rbpr2* is critical for photoreceptor cell viability and visual function under dietary vitamin A restriction.** (**A**) OCT retinal analysis of 12-week old controls (WT, *Rbpr2^fl/fl^*) and homozygous *Rbpr2**^−/−^* mice on vitamin A-deficient diets. Note that the photoreceptor cell layer is reduced (*p* < 0.05) in *Rbpr2^−/−^* mice compared to controls. (**B**) Fundus imaging of 12-week old wild-type (WT), floxed (*Rbpr2^fl/fl^*), and homozygous (*Rbpr2^−/−^*) mice after intraperitoneal injection with ICG (15 mg/kg, Acros Organics, NJ, USA). Note that blood vessel leakage was observed in *Rbpr2^−/−^* mice compared to controls. (**C**–**F**) Scotopic electroretinograms of 12-week old control (*Rbpr2^fl/fl^*) and *Rbpr2**^−/−^* mice on vitamin A-deficient diets, *Rbpr2^−/−^* vs. controls at 0.1 cds/m^2^.

**Figure 7 nutrients-14-02371-f007:**
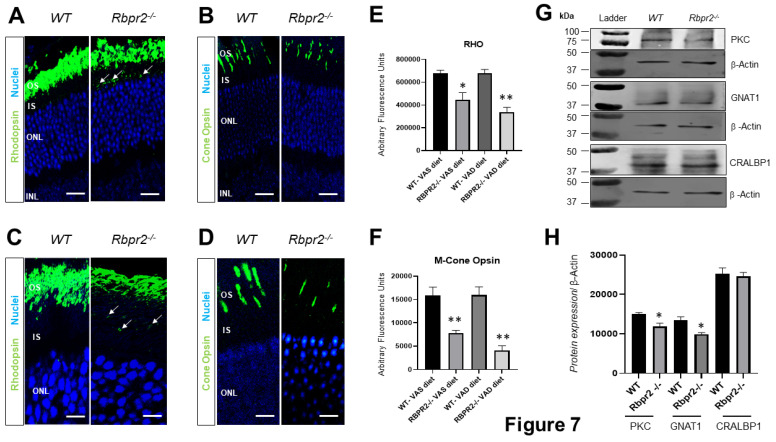
**Immunohistochemical analysis for rhodopsin and cone opsin in retinas of*****Rbpr2******^−/−^*****mice.** Levels and localization of rhodopsin (Rho) in wild-type (WT) and *Rbpr2**^−/−^* mice on either vitamin A-sufficient diets (**A**) or vitamin A-deficient diets (**C**). Red/green medium wavelength cone opsin (M-opsin) staining in WT and *Rbpr2**^−/−^* mice on either vitamin A-sufficient diets (**B**) or vitamin A-deficient diets (**D**). Loss of cone opsin staining was observed in *Rbpr2^−/−^* mice under both dietary conditions (** *p* < 0.005). *Rbpr2^−/−^* mice on either vitamin A diet showed a significant loss of rhodopsin staining (* *p* < 0.05; ** *p* < 0.005) and mislocalization to the inner segments (arrows in panels A and C). Images in panels (**A**–**D**) are representative of immunostained retinal sections (*n* = 5–7 sections per eye) imaged from *n* = 8 animals per genotype and age group (50:50 male to female ratio). (**E**,**F**) Quantification of rhodopsin and cone-opsin in OS among various genotypes and dietary conditions (* *p* < 0.05; ** *p* < 0.005). Scale bars = 75 µm (**A**,**B**). Scale bars = 25 µm (**C**,**D**). OS, outer segments; IS, inner segments; ONL, outer nuclear layer; INL, inner nuclear layer. (**G**) Protein expression analysis of components of the photoreceptors. Western blot analysis for PKC, GNAT1, and CRALBP1 in total protein preparations of the retina in vitamin A-deficient *Rbpr2**^−/−^* mice and WT mice (pooled *n* = 6 retinas per genotype and age). β-actin gene expression was used as the internal control. (**H**) Protein densitometry analysis and quantification. The statistical analysis was performed using an unpaired two-tail Student *t*-test by comparing values from age-matched *Rbpr2**^−/−^* and WT mice. * *p* < 0.05.

**Figure 8 nutrients-14-02371-f008:**
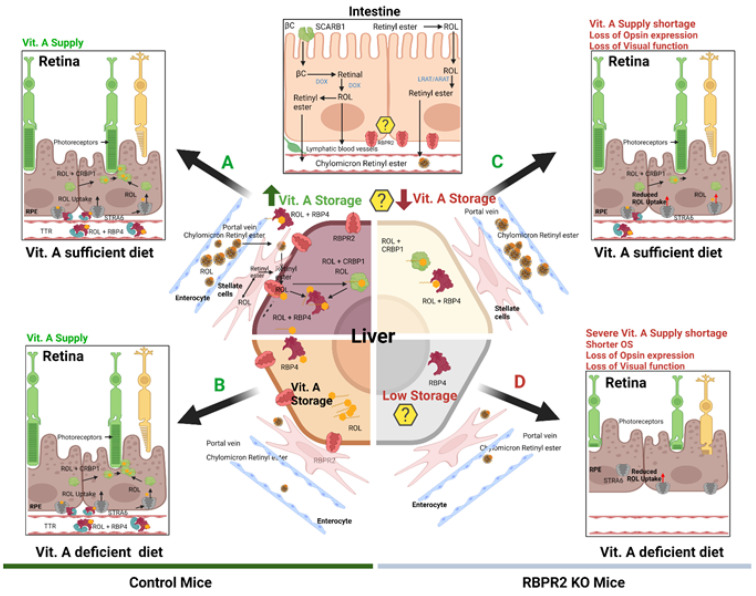
**Schematic representation of the loss of vitamin A transport and its consequences on retinal homeostasis in *Rbpr2^−/−^* mice.** Membrane receptor proteins mediate and recognize functions of RBP4 for ROL uptake. “Stimulated by retinoic acid 6” (STRA6) in the eye and “retinol binding protein receptor 2” (RBPR2, also known as STRA6Like) in liver and intestine are involved in retinol uptake and its coupling to RAR/RXR signaling in target cells. The specific RBP4 receptor in the liver (RBPR2) is thought to also mediate reverse transport of retinol (? Symbols), allowing a cycle of retinol between the circulation and liver. When compared to WT mice, *Rbpr2^−/−^* mice on either vitamin A-sufficient (VAS) or vitamin A-depleted diets (VAD) show reduced ocular retinoid concentrations and loss of visual function. Additionally, *Rbpr2^−/−^* mice show severe retinal phenotypes when challenged with a VAD diet. Thus, the retinal function impairment could be caused by a deficiency in systemic retinol transport to the eye, which is likely dependent on the function of RBPR2. The model was created with BioRender.com (https://help.biorender.com/en/articles/3619405-how-do-i-cite-biorender, accessed on 5 January 2022).

## Data Availability

Not applicable.

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
