# Peer review of "Mice Lacking the Systemic Vitamin A Receptor RBPR2 Show Decreased Ocular Retinoids and Loss of Visual Function"

_nutrients, 2022, doi:10.3390/nu14122371_

Round 1

Reviewer 1 Report

This study provides an initial evaluation on the effect of RBPR2 null mutation on visual function in the young adult mouse and  provides a well detailed examination of changes in photoreceptors through histology, electron microscopy and  OCT analysis.  Some issues arise from the study as outlined below:

  1. The abstract assumes the reader knows that the protein of interest, RBPR2, is not expressed in the eye. From what is discussed in the abstract though, it might be assumed RBPR2 functions there and so becomes confusing when discussed as a systemic transporter.  This would become clearer if it were stated at the beginning that RBPR2 is absent from the eye (assuming that is the case).   A further part of the confusion in the report is that it describes RBPR2 as a “ROL transporter” as opposed to receptor.  A ROL transporter or systemic transporter, would include RBP4 itself.  Given it is retinol-binding protein 4 receptor 2 it would be clearer to describe as a RBP4 receptor.
  2. The abstract describes “under conditions replicating vitamin A excess and deficiency”. Which were the experiments under conditions of excess?
  3. The methods section describes a custom rabbit polyclonal antibody made against mouse RBPR2 that does will not cross-react with mouse STRA6. More details are needed e.g. do the authors know what the light band is in figure 1C, particularly in lung?  Might this be STRA6?
  4. No antigen retrieval is described in the methods for IHC. Is this correct?
  5. Also in the methods section, more information is needed on how “fluorescence within individual retinal layers was quantified using Image J or Fiji”. How were regions to quantify chosen?  It is unlikely there will be a linear response in fluorescence intensity.  Could any other measures be used?  If not, the authors should be more cautious in the statement of a decline in the proteins detected using this (in both results and discussion section).
  6. Figure 2 shows that RBPR2 is barely expressed in the eye, although a very weak band is present in panel A – does this imply e.g. contaminating tissue? Or some small subset of cells expressing RBPR2?
  7. How was the particular timing decided for low vitamin A diets/vitamin A depleted (VAD) diets for 8-weeks?  To what extent would this be expected to deplete body stores of vitamin A?   Comparing figures 3 and 5 there appears to be no changes in occular vitamin A levels in wild type animals. How does this correlate with “humans not actively consuming vitamin A-rich foods”, which reads like a normal human diet.  Supplementary figure S4 shows low detail images of a variety of organs after treatment – would changes be expected to be seen in such images of low resolution?  Details on the vitamin A deficiency diet is needed in the methods section.
  8. The discussion describes how the results support “a physiological role for RBPR2 in the systemic transport of ROL to the eye in support of retinal homeostasis and visual function.” and that retinoid signalling was normal in other tissues, stating “indicated that retinoid signaling in these organs was likely maintained by circulatory retinyl esters in chylomicrons”. However the present results provide insufficient evidence for this, simply showing a gross view of organs other than the eye and single histological sections.  To be confident there are no abnormalities in other organs would require the type of detailed analysis applied to the eye.  Without this statements such as  there is “a more direct role for RBPR2 in the transport of dietary vitamin “ compared to other organs or “mice are phenotypically normal except for their vision.” are not strongly supported.
  9. The statement in the discussion “it is possible that age is a determining factor for the observation of retinal phenotypes, evident only when both systemic and ocular retinoid content get severely depleted as these Rbpr2-/- mice age.” Is unclear. What is hypothesised to occur with increasing age?
  10. What is the significance of increase in vessel leakage in the Rbpr2-/- vitamin A depleted animals? This is mentioned in the figure 6 legend but not discussed elsewhere.  Semi-quantitative western blotting showed a decrease in PKC and GNAT1 expression in vitamin A sufficient diets conditions.  Was there any information in deficient conditions?

Minor

Font sizes were uneven on p12 figure 7 legend

Author Response

Reviewer 1

Comments and Suggestions for Authors

This study provides an initial evaluation on the effect of RBPR2 null mutation on visual function in the young adult mouse and provides a well detailed examination of changes in photoreceptors through histology, electron microscopy and  OCT analysis.  Some issues arise from the study as outlined below:

Author Response: We thank the reviewer for their comments and suggestions and for the support of our manuscript. Please see detailed responses below. All revisions are highlighted in yellow in the revised manuscript.

  1. The abstract assumes the reader knows that the protein of interest, RBPR2, is not expressed in the eye. From what is discussed in the abstract though, it might be assumed RBPR2 functions there and so becomes confusing when discussed as a systemic transporter.  This would become clearer if it were stated at the beginning that RBPR2 is absent from the eye (assuming that is the case).   A further part of the confusion in the report is that it describes RBPR2 as a “ROL transporter” as opposed to receptor.  A ROL transporter or systemic transporter, would include RBP4 itself.  Given it is retinol-binding protein 4 receptor 2 it would be clearer to describe as a RBP4 receptor.

Author Response 1: We thank the reviewer for their comment. To make it clearer for the reader and to understand the previously reported differential expression patterns between STRA6 and RBPR2 in mice, we have added additional information about the RBP4-vitamin A receptor, RBPR2, in the abstract and introduction. We also agree with the reviewer and have replaced the wording “transporter” with “receptor”, where we mention RBPR2 function. This change is now reflected in the title, abstract, introduction, supplementary information, and elsewhere in the text.

  1. The abstract describes “under conditions replicating vitamin A excess and deficiency”. Which were the experiments under conditions of excess?

Author Response 2: We have replaced “excess” with “sufficient” as no experiments were conducted in animals fed a Vitamin A excess diet.

  1. The methods section describes a custom rabbit polyclonal antibody made against mouse RBPR2 that does will not cross-react with mouse STRA6. More details are needed e.g. do the authors know what the light band is in figure 1C, particularly in lung?  Might this be STRA6?

Author Response 3: We have added additional information about the custom peptides used to generate the RBPR2 antibody in the methods and show this information in the new Supplementary Figure S1. The light bands in Figure 1C could be background, as this image was obtained using a longer exposure time.

  1. No antigen retrieval is described in the methods for IHC. Is this correct?

Author Response 4: No antigen retrieval was used in this methodology. This methodology has been used in prior publications by us (Solanki A.K., Cells, 2020; Rohrer B, IJMS, 2021).

  1. Also in the methods section, more information is needed on how “fluorescence within individual retinal layers was quantified using Image J or Fiji”. How were regions to quantify chosen?  It is unlikely there will be a linear response in fluorescence intensity.  Could any other measures be used?  If not, the authors should be more cautious in the statement of a decline in the proteins detected using this (in both results and discussion section).

Author Response 5: Briefly, immunostained retinal images were exported and opened using the Image J software and the free hand tool was used to draw the region of interest (ROI) in the retinal layers. From the Analyze menu, measurements were set to calculate area-integrated intensity and mean gray value. The background values were noted in areas with no fluorescence. Corrected fluorescence was measured as difference of integrated density and area, adjusted for background fluorescence. The quantified values from the multiple ROIs were plotted and then averaged. Statistical significance was measured using t-test, Rbpr2-KO retinas were compared to WT or controls. We have added this information to the methodology section (section 2.7).

  1. Figure 2 shows that RBPR2 is barely expressed in the eye, although a very weak band is present in panel A – does this imply e.g. contaminating tissue? Or some small subset of cells expressing RBPR2?

Author Response 6: We do not observe any RBPR2 expression in the murine eye (Figure 1C). The lack of RBPR2 expression in the murine eye has been reported previously and confirmed by other groups (Alapatt P, 2013, JBC, Fig. 4A).

  1. How was the particular timing decided for low vitamin A diets/vitamin A depleted (VAD) diets for 8-weeks?  To what extent would this be expected to deplete body stores of vitamin A?   Comparing figures 3 and 5 there appears to be no changes in occular vitamin A levels in wild type animals. How does this correlate with “humans not actively consuming vitamin A-rich foods”, which reads like a normal human diet.  Supplementary figure S4 shows low detail images of a variety of organs after treatment – would changes be expected to be seen in such images of low resolution?  Details on the vitamin A deficiency diet is needed in the methods section.

Author Response 7: Based on previous studies in Stra6, Lrat, and Rbp4 deficient mice, we were interested in studying the effects of RBPR2 loss under variable conditions of dietary vitamin A in young animals (10-12 weeks of age). It is know that in WT mice under VAD conditions, ocular retinoid concentrations are maintained, due to stored vitamin A being transported by RBP4 to the eye and taken up by STRA6 (Amengual J, HMG, 2013; Kelly M, FASEB J, 2018). We observed a similar situation in our study with WT mice fed VAD diets (Figure 5B). Supplementary Figure S4 was included to provide the reader with gross pathology and overview of ocular and peripheral organs from Rbpr2-KO mice. We believe it might be of interest to readers, to report such developmental data, as Rbpr2-/- mice like Stra6-/- and Rbp4-/- mice are phenotypically normal, but effects of their loss on internal organ development in these mice have never been shown in publications. Additionally, we have sectioned and stained these tissues to show at the microscopic level the lack of any pathology. We have added details of the VAD diet in the methods (section 2.3).

  1. The discussion describes how the results support “a physiological role for RBPR2 in the systemic transport of ROL to the eye in support of retinal homeostasis and visual function.” and that retinoid signaling was normal in other tissues, stating “indicated that retinoid signaling in these organs was likely maintained by circulatory retinyl esters in chylomicrons”. However the present results provide insufficient evidence for this, simply showing a gross view of organs other than the eye and single histological sections.  To be confident there are no abnormalities in other organs would require the type of detailed analysis applied to the eye.  Without this statements such as  there is “a more direct role for RBPR2 in the transport of dietary vitamin “ compared to other organs or “mice are phenotypically normal except for their vision.” are not strongly supported.

Author Response 8: We agree with this comment and have addressed it in the discussion. Due to COVID we were limited in collaborating with other research groups to perform a more detailed functional analysis of non-ocular organs from Rbpr2-/- mice.

  1. The statement in the discussion “it is possible that age is a determining factor for the observation of retinal phenotypes, evident only when both systemic and ocular retinoid content get severely depleted as these Rbpr2-/- mice age.” Is unclear. What is hypothesised to occur with increasing age?

Author Response 9:  Unlike Stra6 and Lrat mice, the physiological function(s) of RBPR2 for long-term retinoid homeostasis are currently limited. We are also interested in studying the contributing roles of Lrat and Stra6 in concert with RBPR2, in Rbpr2-/- mice, as they age. Additionally, Rbpr2 receptor not only has uptake capabilities for circulatory RBP4-ROL but likely also plays a significant role (with LRAT) in facilitating long-term vitamin A storage in peripheral tissues, as we have shown previously in zebrafish. In a recent study, from the von Lintig group, in Stra6-/- and Isx-/- deficient mice, a greater than 100-fold higher Rbpr2 mRNA expression and increased liver retinoid concentrations was observed in these mutant mice [ref#33]. This indicates that the systemic retinol receptor, RBPR2, is physiologically involved in the reuptake of RBP4-ROL from the circulation, to prevent toxic accumulation of excessive amounts of ROL in these mutant mice [33]. Therefore, this is a general hypothesis and we are currently investigating this in older mice (6-18 months of age). We have added this point on page 15 of the discussion.

  1. What is the significance of increase in vessel leakage in the Rbpr2-/- vitamin A depleted animals? This is mentioned in the figure 6 legend but not discussed elsewhere.  Semi-quantitative western blotting showed a decrease in PKC and GNAT1 expression in vitamin A sufficient diets conditions.  Was there any information in deficient conditions?

Author Response 10: Atrophy of the choroid and neovascularization has been previously observed in Stra6-/- deficient mice and patients with RBP4 mutations (Refs # 12 and #70; Amengual J et al, HMG, 2014 and Seeliger MW et al, IOVS, 1999),  eluding to the possible decrease(s) in ocular retinoid concentrations as a contributing factor. Given this, we were interested in investigating if this pathology was also the case in Rbpr2-/- mice fed VAS or VAD diets. We have added this information to section 3.5. Figure 7E: This WB analysis was from mice fed a VAD diet. WB analysis was performed from total protein preparations of the retinas isolated from Rbpr2-/- mice and WT mice fed a VAD diet. We did not perform this analysis on mice fed a VAS diet.

Minor

Font sizes were uneven on p12 figure 7 legend

Author Response: we have correct this.

Reviewer 2 Report

The authors presented a mouse model of RBPR2 deletion that manifested functional and histological changes in the retina under Vitamin A deficient conditions. Although the findings are intriguing and potentially important, the manuscript requires significant revision.

  1. Vitamin A replete diet = VAD diet? It is confusing.
  2. ONL thickness measurement (nuclei count) from histology was mentioned in methods section, but there was only OCT results in the paper, and the quantification method of OCT images was absent. Although outer segment was clearly shortened in mutant mice on VAD diet, figure 4B showed no ONL thinning which called into question about the accuracy of OCT measurements.
  3. Since cone cell distribution in mouse retina may not be uniform, quantification of M-opsin in sections alone could be misleading.
  4. Retinoid levels presented in this paper were relative values, only comparable among themselves. It is impossible to validate. Please provide absolute values. HPLC data presented RE and atROL separately, one cannot tell how the statistics was done, on which retinoid or which diet.
  5. ERG graphs in figure 3 had no Y axis label and wrong X axis label (the X axis values looked be in log units). ERG data in figure 6 need to be sufficiently described. P value was given, yet no one could tell if it came from a-wave or b-wave amplitude.
  6. ICG angiogram 10mins after ip injection has too many systemic variables to be the sole evidence of leaky retinal vessels.
  7. Figure 7 legend about rhodopsin contradicted what was in the text on page 10. It was strange to see images of the same staining with different magnifications in the same figure. How was immuno-histology and Western quantified?
  8. What were the retinoid levels in liver and blood of these mutant mice?

Author Response

Reviewer #2

Comments and Suggestions for Authors

The authors presented a mouse model of RBPR2 deletion that manifested functional and histological changes in the retina under Vitamin A deficient conditions. Although the findings are intriguing and potentially important, the manuscript requires significant revision.

Author Response: We thank the reviewer for their comments and suggestions and for the support of our manuscript. Please see detailed responses below. All revisions are highlighted in yellow in the revised manuscript.

  1. Vitamin A replete diet = VAD diet? It is confusing

Author Response 11: To be consistent with the literature, we have replaced the word “replete” with “deficient” in the revised manuscript.

  1. ONL thickness measurement (nuclei count) from histology was mentioned in methods section, but there was only OCT results in the paper, and the quantification method of OCT images was absent. Although outer segment was clearly shortened in mutant mice on VAD diet, figure 4B showed no ONL thinning which called into question about the accuracy of OCT measurements.

Author Response 12: We have included OCT measurements in Supplementary Figure S6, while there was a significant decrease in photoreceptor cell layer (PRL) thickness in animals fed the VAD diets, we only observed a trend in ONL thickness decrease in Rbpr2-/- animals fed the VAD diets.

  1. Since cone cell distribution in mouse retina may not be uniform, quantification of M-opsin in sections alone could be misleading.

Author Response 13: We agree and have planned to perform additional cone-opsin analysis in Rbpr2-/- mice. We have also indicated this limitation in our revised manuscript (discussion page 16).

  1. Retinoid levels presented in this paper were relative values, only comparable among themselves. It is impossible to validate. Please provide absolute values. HPLC data presented RE and atROL separately, one cannot tell how the statistics was done, on which retinoid or which diet.

Author Response 14: We have now presented absolute (Total ocular retinoid concentrations in pmole/eye) in Figure 3 (animals fed VAS diets) and Figure 5 (animals fed VAD diets). Absorbance is presented in Supplementary Figure S3A and S3D.

  1. ERG graphs in figure 3 had no Y axis label and wrong X axis label (the X axis values looked be in log units). ERG data in figure 6 need to be sufficiently described. P value was given, yet no one could tell if it came from a-wave or b-wave amplitude.

Author Response 15: we have corrected these figures. We have indicated that P value came independently from a- and b- wave (Figure legends 3 and 6).

  1. ICG angiogram 10mins after ip injection has too many systemic variables to be the sole evidence of leaky retinal vessels.

Author Response 16: Atrophy of the choroid and neovascularization has been previously observed in Stra6-/- deficient mice and patients with RBP4 mutations (Refs # 12 and #70; Amengual J et al, HMG, 2014 and Seeliger MW et al, IOVS, 1999),  eluding to the possible decrease(s) in ocular retinoid concentrations as a contributing factor. Given this, we were interested in investigating if this pathology was also the case in Rbpr2-/- mice fed VAS or VAD diets. We have added this information to section 3.5.

  1. Figure 7 legend about rhodopsin contradicted what was in the text on page 10. It was strange to see images of the same staining with different magnifications in the same figure. How was immuno-histology and Western quantified

Author Response 17: we apologize for this oversight and have made the necessary corrections (page 12). Post IF staining, retinas were imaged either using a Keyence microscope or a Lecia confocal microscope. IF imaging and quantification and WB protein quantification methods are indicated in the methodology (Section 2.7 and Section 2.14).

  1. What were the retinoid levels in liver and blood of these mutant mice?

Author Response 18: We have shown liver ROL and Serum ROL levels in Supplementary Figure S3B and S3C, which were lower in Rbpr2-/- mice compared to controls, on VAS diets.

Round 2

Reviewer 1 Report

The earlier critique was made " The discussion describes how the results support “a physiological role for RBPR2 in the systemic transport of ROL to the eye in support of retinal homeostasis and visual function.” and that retinoid signaling was normal in other tissues, stating “indicated that retinoid signaling in these organs was likely maintained by circulatory retinyl esters in chylomicrons”. However the present results provide insufficient evidence for this, simply showing a gross view of organs other than the eye and single histological sections. To be confident there are no abnormalities in other organs would require the type of detailed analysis applied to the eye. Without this statements such as there is “a more direct role for RBPR2 in the transport of dietary vitamin “ compared to other organs or “mice are phenotypically normal except for their vision.” are not strongly supported."

The authors agreed but all these strong statements remain.  The present results may simply support the fact that the eye has a very high demand for vitamin A and so when vitamin A delivery is depleted the greatest effect is on this tissue.  There may be more minor effects on other tissues with lesser vitamin A demand that are not evident using the approaches used in this study.  To say, e.g. “mice are phenotypically normal except for their vision.” is not fully supported by this study.

Author Response

We thank the reviewer for their helpful comments and suggestions. We agree and have included this statement (pgs. 587-592).